# NSCLC Biomarkers to Predict Response to Immunotherapy with Checkpoint Inhibitors (ICI): From the Cells to In Vivo Images

**DOI:** 10.3390/cancers13184543

**Published:** 2021-09-10

**Authors:** Virginia Liberini, Annapaola Mariniello, Luisella Righi, Martina Capozza, Marco Donatello Delcuratolo, Enzo Terreno, Mohsen Farsad, Marco Volante, Silvia Novello, Désirée Deandreis

**Affiliations:** 1Department of Medical Science, Division of Nuclear Medicine, University of Turin, 10126 Turin, Italy; desiree.deandreis@unito.it; 2Nuclear Medicine Department, S. Croce e Carle Hospital, 12100 Cuneo, Italy; 3Thoracic Oncology Unit, Department of Oncology, S. Luigi Gonzaga Hospital, University of Turin, 10043 Orbassano, Italy; annapaola.mariniello@unito.it (A.M.); dona92m@tiscali.it (M.D.D.); silvia.novello@unito.it (S.N.); 4Pathology Unit, Department of Oncology, S. Luigi Gonzaga Hospital, University of Turin, 10043 Orbassano, Italy; luisella.righi@unito.it (L.R.); marco.volante@unito.it (M.V.); 5Molecular & Preclinical Imaging Centers, Department of Molecular Biotechnology and Health Sciences, University of Torino, Via Nizza 52, 10126 Torino, Italy; martina.capozza@unito.it (M.C.); enzo.terreno@unito.it (E.T.); 6Nuclear Medicine, Central Hospital Bolzano, 39100 Bolzano, Italy; mohsen.farsad@sabes.it

**Keywords:** immune checkpoint inhibitors, non-small cell lung carcinoma, PD-1, PD-L1, immune PET, immunotherapy, radiomics, PET/CT

## Abstract

**Simple Summary:**

Lung cancer and in particular non-small cell lung cancer (NSCLC) remains the leading cause of cancer-related death. The development of new therapeutic approaches, including immunotherapy, has led to substantial improvement in survival time and quality of life. However, the clinical benefit of immunotherapy-based strategies is still limited to a minority of patients, reflecting the need to identify predictive biomarkers of response, which are any substance, structure, or process or its products that can be measured in the body and that can influence or predict clinical response. In this work, we provide an overview of the approved and the most promising investigational biomarkers, which have been assessed in vitro/ex vivo and in vivo, to identify patients who could benefit the most from immunotherapy-based treatment.

**Abstract:**

Lung cancer remains the leading cause of cancer-related death, and it is usually diagnosed in advanced stages (stage III or IV). Recently, the availability of targeted strategies and of immunotherapy with checkpoint inhibitors (ICI) has favorably changed patient prognosis. Treatment outcome is closely related to tumor biology and interaction with the tumor immune microenvironment (TME). While the response in molecular targeted therapies relies on the presence of specific genetic alterations in tumor cells, accurate ICI biomarkers of response are lacking, and clinical outcome likely depends on multiple factors that are both host and tumor-related. This paper is an overview of the ongoing research on predictive factors both from in vitro/ex vivo analysis (ranging from conventional pathology to molecular biology) and in vivo analysis, where molecular imaging is showing an exponential growth and use due to technological advancements and to the new bioinformatics approaches applied to image analyses that allow the recovery of specific features in specific tumor subclones.

## 1. Introduction

Lung cancer is one of the most common tumors and is the leading cause of cancer-related death worldwide, accounting for about 2.21 million new cases and 1.80 million deaths in 2020 [1,2]. The majority of lung cancers (80–90%) are non-small cell lung cancers (NSCLC), which are most commonly diagnosed at advanced stages of the disease (65%), with an estimated 5-year overall survival rate of 18% [3,4].

However, the recent availability of targeted strategies and of immunotherapy with immune-checkpoint inhibitors (ICI) is favorably changing prognosis. Indeed, the treatment of advanced stage NSCLC has evolved from the empiric use of chemotherapy to a armamentarium of tailored approaches, with subsets of patients treated according to the genetic alterations of their tumor and the status of programmed death ligand-1 (PD-L1), which predict the response to targeted therapies or immune checkpoint inhibitors, respectively [4].

Only 25% of NSCLC are oncogene-addicted, showing that genetic mutations are targetable with highly selective tyrosine kinase inhibitors [5]. To date, in the remaining cases, which account for the large majority of NSCLC patients, no actionable driver mutations are known, and prognosis is usually poorer [1,2]. In recent years, immunotherapy with ICI has provided substantial survival benefits across all lines of treatment [6].

Immune checkpoints proteins (ICs), such as cytotoxic T-lymphocyte associated protein 4 (CTLA-4) and programmed cell death protein 1 (PD-1) are co-inhibitory receptors present on activated T cells [7,8,9,10,11,12,13]. Overexpression of PD-1, a transmembrane protein receptor of the CD28 family, is a hallmark of T cell exhaustion, a process characterized by the functional impairment of antigen-experienced CD8^+^ T cells [14,15]. PD-1 has two ligands, both members of the B7 family: PD-L1 and PD-L2. PD-L1 is constitutively expressed by many different cells of hematopoietic and non-hematopoietic origin and can be induced by type I and II IFN and other cytokines [16]. The upregulation of PD-L1 in response to IFN-γ has been described as a resistance mechanism to dampen ongoing immune responses and to protect healthy tissues [17].

Cancer cells have learned to take advantage of the PD-L1 overexpression to evade immune surveillance. In fact, when the PD-1 on activated CD8^+^ T cells interacts with its corresponding ligands on antigen presenting cells and/or tumor cells, T cell receptor (TCR) signaling is attenuated, resulting in reduced T cell effector functions in a process that ultimately generates exhaustion [11]. In both chronic infections and in human cancer therapeutics, the blockade of PD-1/PD-L1 interaction with monoclonal antibodies has been demonstrated to reinvigorate exhausted CD8^+^ T cells, restoring T cell proliferation and the production of effector molecules (perforin, granzymes and cytokines) [14,18]. PD-L1 expression in tumor cells/microenvironment has been correlated to response to treatment with the PD-1 inhibitor and to date represents the only available biomarker of response for single agent PD-1/PD-L1 blockade [19].

CTLA-4 is the second IC used in clinical practice for NSCLC (USA only). It plays a central role in T cell priming and early activation in lymphoid organs. The inhibitory signal on TCR, generated upon interaction with its ligand CD80/86, for which it competes with CD28, downregulates antigen-specific T cell expansion [20]. In addition, CTLA-4 is not only expressed on activated T effector cells but is also constitutively exposed on T regulatory cells. This suggests that the CTLA-4 blockade mostly exerts its anti-cancer activity by suppressing this inhibitory CD4^+^ T cell population [21,22]. On the other hand, the direct effects of anti-CTLA-4 in increasing effector T cell functionality is still not completely clear in humans.

ICI with PD-1/PD-L1 blockade represents the current backbone of systemic treatment strategies for advanced non-oncogene addicted NSCLC [6,23]. The choice to use PD-1/PD-L1 blockade alone or in combination with other agents (chemotherapy/chemotherapy plus anti-angiogenetics/anti-CTLA4) is based on the line of treatment and PD-L1 expression on tumor cells [24], as shown in Table 1.

However, despite the undiscussed survival advantage demonstrated in phase III clinical trials, The benefit of ICIs is not universal, and only a minority of patients experience a durable response. Moreover, unpredictable paradoxical responses in terms of expansive tumor growth after ICI have also been described and are collectively known as disease hyperprogression [37]. These indicate that primary or acquired resistance to ICI are due, at least in part, to poor patient selection and a knowledge gap in the biological underpinnings.

Accumulating evidence has shown that PD-L1 assayed with immunohistochemistry alone is not accurate or reliable enough to predict response to ICIs [38,39], and currently, the identification of alternative/companion biomarkers is a research priority. What is more likely is that response to ICIs may depend on a set of factors, both host- and tumor-related, that are involved in the complex immune anti-tumor control network [38,39].

Here, we provide an overview on both in vitro and in vivo biomarkers (as sketched in Figure 1) that are already in use or under development and that may enrich clinical benefit of ICI treatment in NSCLC patient candidates.

We searched the PubMed, PMC, Scopus, Google Scholar, Embase, Web of Science, and Cochrane library databases (between January 2010 and July 2021), using the following terms (as text and as MeSH terms): “Immunotherapy”, “ICI*”, “response biomarkers”, “predictive factors”, “anti-PD-1”, “anti-PD-L1”; “anti-CTLA-4”; “nivolumab”, “pembrolizumab”, “durvalumab”, “cemiplimab”, “atezolizumab”, “ipilimumab”; “Positron-Emission Tomography”, “Positron Emission Tomography Computed Tomography”, “PET/MR*”, “PET/CT”, “Lung Neoplasms”, “Carcinoma, Non-Small-Cell Lung”, “ICOS”, “immune PET”, “immune-PET”, “ICI*”, “Fluorodeoxyglucose F18”, “radiomic*”, and “Artificial Intelligence”.

No language restriction was applied to the search, but only articles in English were reviewed. The systematic literature search returned 565 articles. According to the PRISMA flow-chart, after duplicate removal, 182 articles were considered, fully read, analyzed, and extensively described according to their title and abstract, as previously described [40]. We also checked for further relevant articles in the references of the articles included in the retrieved literature.

## 2. In Vitro Biomarkers for Immunotherapy

The search for indicators of clinical benefit after ICI is currently a hot topic and involves various fields of knowledge, from conventional pathology to molecular biology. Due to the technological advancement in genetic sequencing platforms and the availability of bioinformatic tools to interpret complex sets of data, putative response biomarkers have been explored in tumor cells, blood, and healthy tissues, at both the cellular and molecular levels. Here, we reviewed approved and investigational predictive factors of response, which were grouped according to their relationship with tumor cells, the immune microenvironment, and host-intrinsic variables.

### 2.1. Tumor-Related Biomarkers

#### 2.1.1. PD-L1 TPS (Tumor Proportion Score)

Consistent with the current knowledge of the PD-1/PD-L1 blockade’s mechanism of action, PD-L1 expression was the first response biomarker developed, and, so far, the most successful guiding treatment choice in locally advanced and metastatic non-oncogene addicted NSCLC [6,23].

Robust clinical evidence from phase III trials support the ability of PD-L1 status in enriching for clinical benefit [20,41,42]. However, PD-L1 TPS (i.e., the percentage of tumor cells with positive membrane staining over the whole amount of tumor cells in the specimen) assessment with immunohistochemistry (IHC) on tumor specimens presents several limitations. Mechanistically, PD-L1 expression on epithelial cells is a sign of an inflamed microenvironment, mainly induced by IFN-gamma, which is produced by surrounding dendritic cells and activated T lymphocytes [11,43]. Hence, by nature, its expression in tissue is highly variable in time and space, even in the context of the same tumor lesion.

Moreover, the predictive role of PD-L1 might also be influenced by tumor histology. Phase III trials testing nivolumab upon chemotherapy failure showed a correlation between PD-L1 expression and clinical benefit in lung adenocarcinoma, whereas in squamous NSCLC, it was neither prognostic nor predictive of response [41,42].

Beyond this intrinsic limitation, some technical issues also affect PD-L1 reliability as a single biomarker. Indeed, each of the approved ICI has been accompanied by a corresponding IHC kit to assess PD-L1 expression. Potential differences in the staining properties and in the use of various cutoff points for expression have long been a matter of debate.

With the aim to harmonize results and clarify the relationship across these different assays, the Blueprint project compared the four assays that are the most commonly used (22C3 for pembrolizumab, 28-8 for nivolumab, SP142 for atezolizumab, and SP263 for durvalumab) [44]. When compared to the other three assays, which showed a similar analytical performance, SP142 was found to be less sensitive. However, it must be specified that the SP421 criteria for positivity differ from those of the other assays for detecting membrane PD-L1 not only in tumor cells but also in tumor infiltrating immune cells [44]. In this respect, the study by Herbst et al. is noteworthy, which showed that PD-L1 scoring on both tumor and immune cells in the tumor microenvironment may be more accurate in predicting response compared to the sole PD-L1 assessment on tumor cells [45]. Among other relevant findings, this study, which was conducted in different tumor types, also demonstrated that further variables related to T cell biology, such as T-helper type 1 gene expression (interferon-gamma, CD27, CXCR3, CD45RO, GZMB and CD8A), CTLA4 expression, and the absence of fractalkine (CX3CL1), in baseline tumor specimens can significantly predict benefit after ICI. With regard to these latter factors, as opposed to what observed for other tumors such as melanoma, their association with response is not always significant in NSCLC.

All things considered, PD-L1 expression is a weak biomarker when used alone, and the search for alternative biomarkers for use in a composite evaluation is underway.

#### 2.1.2. Tumor Mutational Burden

With the advent of novel bioinformatics tools, growing attention has been dedicated to biomarkers related to cancer neoantigen load [46]. Tumor mutational burden (TMB) is defined as the number of non-synonymous mutations within the coding portion of the cancer genome and possibly reflects in a higher source of neoantigens able to stimulate anti-cancer T cell response. When assessed with targeted Next Generation Sequencing (NGS) platforms, TMB can be considered as the best surrogate available for neoantigen load in terms of applicability on a large scale [47,48].

Seminal studies applied whole exome sequencing to describe and highlight the predictive role of TMB in response to ICIs [48,49,50]. However, due to the high costs, long turnaround time, and needed expertise, targeted panels have been used on both tissue and plasma specimens, demonstrating a close correlation with whole-exome sequencing [51,52]. Similarly, concerning PD-L1 testing, different cut-offs and panels have been used according to the ICIs that have been tested. A further issue concerns lung cancer tissue availability, which is often scarce in clinical practice and is insufficient for TMB evaluation on tissue. On the other hand, when TMB is tested on blood, it is the amount of circulating DNA is critical.

As a newer candidate biomarker, compared to PD-L1 TPS, results from prospective validation are scarce and are often inconsistent. The Checkmate 227 trial established a clinically meaningful TMB cut-off (≥10 mutations per megabase, using the FoundationOne CDx assay) for the progression free survival (PFS) benefit after double ICIs with nivolumab plus ipilimumab vs. chemotherapy. Notably, PFS benefit in high TMB patients occurred irrespectively of PD-L1 expression [53]. However, when the same threshold was used to discriminate overall survival (OS) benefit upon the same regimen, TMB failed to achieve significant results over chemotherapy [54]. Likewise, the NEPTUNE trial, which prospectively assessed response to double ICIs (durvalumab plus tremelimumab) according to the TMB level in blood, failed to demonstrate longer OS in patients with high TMB (≥20 mutations per megabase) [55]. Based on these data, the fact that the performance of tissue and blood TMB still needs to be optimized cannot be ignored before entering clinical practice.

Despite this limitation, based on the overall response rates from a pre-planned retrospective analysis from the KEYNOTE-158 trial, in June 2020, the FDA granted approval for the use of pembrolizumab in patients affected by advanced solid tumors with high tissue TMB (≥10 mutations per megabase evaluable with the FoundationOneCDx or the MSK-IMPACT assays) upon the failure of first-line chemotherapy [56].

Interestingly, a refined version of TMB has emerged from a large meta-analysis on patients receiving ICI for various cancer types, including 76 NSCLC. By analyzing tumor and immune cells, it showed that clonal TMB (i.e., the number of non-synonymous mutations estimated to be present in every cancer cell) outperforms TMB as a whole in predicting sensitization to ICI [57]. Importantly, this work, which utilized standardized bioinformatics workflows and clinical outcome criteria, underlines the potential of multivariable prediction models integrating tumor- and T cell-intrinsic mechanisms of response. As a limitation, it must be considered that these translational works are generally retrospective and are designed with objective response as the endpoint for clinical outcome instead of survival benefit.

An alternative approach to enhance the performance of blood TMB could be the adjustment for allele frequency, as proposed by a retrospective report from the POPLAR and OAK trials. Here, low allele frequency blood TMB (mutation counts with an allele frequency < 5%) significantly predicted favorable OS, PFS, and objective response rate [58].

Another strategy to optimize tissue TMB performance integrates tumor mutations count with epigenetic alterations, showing that high TMB is more frequently associated to aberrant DNA methylation [59]. However, both approaches need prospective validation.

#### 2.1.3. Tumor Genotype

Whether it is well-acknowledged that actionable mutations in the driver oncogenes epidermal growth factor receptor (*EGFR*) and anaplastic lymphoma kinase (*ALK*) confer resistance to ICI [60,61], accumulating evidence suggests a possible positive or negative predictive role for other tumor mutations, involved in various cancer pathways, that may improve the predictive ability of both PD-L1 and TMB when combined.

Immunotarget retrospective analysis evaluated response to ICI therapy in metastatic NSCLC patients with oncogenic driver alterations including Kirsten Rat Sarcoma virus (*KRAS*), *EGFR*, v-raf murine sarcoma viral oncogene homolog B1 (*BRAF*), human epidermal growth factor receptor 2 (*HER2*), mesenchymal epithelial transition factor (*MET*), mutations, and *ALK*, c-ros oncogene 1 (*ROS1*) and rearranged during transfection (*RET*) rearrangements. Evaluating the 12-month progression free survival (PFS), the best outcome to ICI therapy was found in NSCLC patients with *KRAS* and *MET* mutations, who achieved the highest 12-month PFS (25.6% and 23.4%, respectively); NSCLC patients with the *HER2* mutation exhibited an intermediate outcome (18.0%), and those with the *RET*, *EGFR*, *ALK,* and *ROS1* mutations exhibited the poorest 12-month PFS outcomes (7.0%, 6.4%, 5.9%, and unevaluable due to small numbers, respectively) [62].

In a recent report, Aggarwal et al. [63] evaluated baseline blood TMB and specific genetic mutations using a 500-gene NGS panel in 66 NSCLC patients receiving first-line pembrolizumab-based treatment. The combination of blood TMB ≥ 16 and the absence of negative predictor mutations, such as *STK11/KEAP1/PTEN* and *HER2* exon 20, was associated with PFS and overall survival (OS) benefit.

Another analysis by Lamberti et al. [64] examined clinical, pathological, and genomic data associated with high (≥50%), low (1–49%) and negative (<1%) PD-L1 expression in 909 non-squamous NSCLC samples. Among other factors, compared to high PD-L1 tumors, negative PD-L1 tumors presented more often *STK11* (19% vs. 5%; *p* < 0.001), *EGFR* (22% vs. 11%; *p* < 0.001), *CTNNB1* (4% vs. 0.4%; *p* < 0.001), and *APC* mutations (5% vs. 1%; *p* = 0.005), respectively. In a smaller subset receiving ICIs, the *STK11* or *EGFR* mutations were also significantly associated with shorter PFS and OS.

A similar study by Schoenfeld et al. [65] on 1586 lung adenocarcinoma patients with paired PD-L1 and targeted-NGS data not only confirmed the association of the *EGFR* and *STK11* mutations with PD-L1 negativity but also showed that the *KRAS*, *TP53*, and *MET* mutations were significantly associated with high PD-L1 expression.

Furthermore, an exploratory analysis from the KEYNOTE-042 trial presented at the AACR 2020 assessed the prevalence of the *STK11* and *KEAP1* mutations and their association with pembrolizumab efficacy over chemotherapy [66]. Despite confirming that *STK11* mutations occurred more often in low PD-L1, a higher frequency has been reported in high TMB patients. More importantly, response rate, PFS, and survival with pembrolizumab were similar irrespective of *STK11* or *KEAP1* mutation, while chemotherapy efficacy was lower in *STK11* mutation carriers.

As for the *KRAS* mutations, which account for 20–30% of lung adenocarcinoma, these represent a heterogeneous category, with specific mutations having different responses to checkpoint inhibition [67].

An example, despite being described in a different tumor type (pancreatic cancer), has been provided for the *KRAS^G12D^* mutation, which has been associated with production of granulocyte–macrophage colony-stimulating factor (GM-CSF), which is involved in the recruitment of myeloid cells, thus maintaining an immunosuppressive environment. On the other hand, it is still unexplored if other *KRAS* single-nucleotide point mutations at exon 12 or 13 (G12V, G12A, G12C and G13D) also mediate GM-CSF production or if this is specific to the G12D variant [68].

Other translational studies have shown an increased immunogenicity of *KRAS*-mutant tumors [69,70]. However, it has been suggested that co-occurring mutations may flip the potential immune stimulating role of *KRAS*. Skoulidis et al. [71] retrospectively assessed 17 *KRAS* mutant NSCLC treated with nivolumab, showing poorer responses in cases of *KRAS/STK11* co-mutation compared to those with *KRAS/TP53* co-mutation. Moreover, in the same report, the association of *STK11* mutation with worse clinical outcome was also confirmed among the PD-L1 positive subgroup of *KRAS/STK11* mutant NSCLC patients treated with ICIs. Mechanistic evidence explaining how *STK11/LKB1* aberration could mediate resistance to ICI has been shown by Koyama et al. in a preclinical study. In a mouse model of *KRAS*-driven NSCLC, genetic ablation of *STK11/LKB1* resulted in the accumulation of neutrophils with T cell-suppressive effects. Neutrophil-depleting antibodies yielded therapeutic benefits associated with reduced neutrophil accumulation and proinflammatory cytokine expression [72].

Favorable effects on ICI outcome have been reported also for NSCLC harboring the A*RID1A* mutation. In the context of double ICIs, an exploratory analysis from the MYSTIC trial evaluated the relationship between gene alterations detected on circulating free DNA and outcome to durvalumab plus tremelimumab treatment [73]. While mutations in the tumor suppressor genes *KEAP1* and *STK11* were found to be associated with poorer prognosis, regardless of the treatment received (chemotherapy or ICI), mutations in the *ARID1A* mutation were only associated with longer survival in patients receiving durvalumab plus tremelimumab. These findings suggest that while the highly common mutations in *KEAP1* and *STK11* may have negative prognostic value in NSCLC, *ARID1A* may deserve further attention as a possible predictive factor of response. In addition, A*RID1A* was retrospectively investigated in 3403 patients affected by 9 tumor types, including NSCLC [74]. *ARID1A* mutations were significantly associated with microsatellite instability and high TMB. Of note, its favorable predictive role on PFS occurred independently from microsatellite or TMB status.

Among others, further mutant genes described as potential predictors of ICI outcome are *POLE* (presence) and *PTEN* (absence); the first is involved in DNA replication and repairs, and the second is an acknowledged tumor suppressor, inhibiting the PI3K/mTOR/AKT pathway [75,76].

Despite being promising and possibly easy to assess, the concept that single tumor mutations, or even a cluster, may predict clinical outcome to ICI needs further prospective and mechanistic studies.

### 2.2. Biomarkers Related to Tumor Microenvironment (TME)

#### 2.2.1. T Lymphocyte Infiltration

Whether high levels of tumor-infiltrating T lymphocytes (both CD3^+^CD8^+^ and CD3^+^CD4^+^) in TME have long been recognized to have a favorable prognostic role [77,78], emerging insight into T cell biology indicates that the rescue of T cell homing at the tumor site is also essential for ICI efficacy.

Undoubtedly, T cell infiltration is a sign of immune recognition and of an inflamed microenvironment.

Activated CD8^+^ T cells are the major actor involved in anti-cancer immunity and are poised to recognize and selectively eliminate host cells expressing intracellular non-self or mutated-self antigens arising from infection or cancer [11].

A recent multiomics analysis estimated CD8^+^ T cell abundance as the most predictive factor of response to anti-PD-1/PD-L1 therapy across 21 cancer types, including NSCLC [79].

Indeed, in the study by Herbst et al. [45], PD-L1 expression on immune cells in TME, including in myeloid cells (macrophages, dendritic cells) and T cells, seemed to better correlate with ICI response than expression on tumor cells, as mentioned in the PD-L1 relative section.

Even though standard hematoxylin and eosin (H&E) staining may be sufficient to assess lymphocyte infiltration, IHC is the optimal tool, allowing the characterization of lymphocyte populations using specific markers [38].

The role of CD4^+^ T cells is more controversial since these are a complex population, with either immune activating (T helpers 1 and 2) or suppressive features (T helper 17, T regulatory cells), thus requiring supplemental markers for association. For example, FOXP3 and CD25 are useful for identifying T regulatory cells [80], the infiltration of which has been associated with poor prognosis in NSCLC [81]. Deeper insight should be gained into the potential functions of T regulatory cells in influencing response to ICI.

A practical limitation for the histological characterization T lymphocyte infiltration in NSCLC clinical practice is the scarce availability of tumor tissue. Indeed, in contrast to other tumor types that are more often amenable to surgical resection, in the majority of cases, NSCLC is diagnosed at advanced stages of disease, when radical approaches are not feasible and when the tumor tissue is limited to cytological samples from small biopsies.

#### 2.2.2. Tumor Infiltrating CD8^+^ T Cell: Phenotype and TCR Clonality

The incorporation of novel sequencing and sorting strategies in laboratory and translational research has allowed the augmentation of the resolution power on T cells in terms of the discovery of new phenotypic markers and evolution tracking. Consequently, this progress in technology has deepened our comprehension on how T cells respond to ICI and set the basis for the development of promising response biomarkers.

Comparing transcriptional, metabolic, and functional signatures of intra-tumoral CD8^+^ T lymphocytes, based on different expression levels of PD-1, Thommen et al. [82] found that CD8^+^ T cells expressing the highest levels of PD-1 displayed the strongest reactivity against tumor cells. Moreover, compatibly with their exhausted state, PD-1^high^ CD8^+^ T cells showed an impaired ability in cytotoxic cytokine production and produced CXCL13, which stimulated immune recruitment at the tertiary lymphoid structure. Importantly, the presence of this phenotype was strongly predictive for both response and survival in a small cohort of NSCLC patients treated with PD-1 blockade.

Litchfield et al. [57] recently confirmed the relevant favorable role of CXCL13 expressed by CD8^+^ T cells on ICI response. This group performed deep transcriptome sequencing with a single-cell RNA of tumor infiltrating CD8^+^ T cells reactive to a clonal cancer neoantigen (MTFR2) in a NSCLC patient and found that CXCL13 was upregulated. In the same study, CXCL13 exhibited the most marked selective expressions in ICI responders across 1008 patients receiving ICI for different tumor types.

Single-cell RNA sequencing is particularly used for the analysis of T cell receptor (TCR) sequencing, thus enabling an indirect and quantitative characterization of antigen-specific T cells.

The clonal architecture of the intra-tumoral or peripheral T cell repertoire can be considered as an indicator of tumor immunogenicity, and its role in predicting ICI clinical benefit is under investigation. Diversity in TCR composition is usually quantified as for richness and clonality. To simplify, richness indicates the number of specific TCR sequences, whereas clonality takes into account the distribution of the frequencies of sequences. Clonality is defined as the complement of evenness (i.e., 1–evenness), whereas high TCR clonality indicates an unequal sequence distribution with the predominance of one or more oligoclonal populations [47,83]. Most of the available evidence derives from melanoma [17,83], reflecting, at least in part, the tissue availability constrains in NSCLC clinical practice mentioned above. Nevertheless, a thorough translational analysis on resected NSCLC specimens from a phase II trial testing nivolumab in the neoadjuvant setting also explored the dynamics of intra-tumoral and peripheral neoantigen-specific T cell clonotypes. Tumors presenting major pathological response after resection showed a higher TCR clonality compared to tumors without a pathological response. Notably, the T cell clones occurring at a higher frequency after surgery were shared between the blood and tissue, and many of these clones were not detected in the peripheral blood before treatment, suggesting that ICIs may induce the expansion of specific T cell clones [84].

However, studies in other tumor types and treatment settings showed conflicting results both for the positive or negative predictive role of high TCR clonality on ICI outcome [85,86,87], indicating that this evidence is very preliminary. Additionally, it must be considered that TCR sequencing is an expensive technique and requires a relatively long time and a high-level of expertise. So far, more work is needed to clarify the utility of T cell diversity as a response biomarker.

#### 2.2.3. Vascularization in Tumor Microenvironment

T cell infiltration in a tumor is mediated by the co-operation of a complex network involving numerous cell types and cytokines. In contrast to what occurs for an infection site, T cell entry into the TME is physically restrained by vascular architecture, which is irregular in tumors, because of the high interstitial fluid pressure and altered expression of adhesion molecules [88]. These characteristics, besides limiting the efficacy of ICIs, have also discouraged their clinical development of T cell adoptive therapies in solid tumors, including chimeric antigen receptor (CAR) T cell therapies [89].

Response biomarkers to ICIs related to neo-angiogenesis, in either tumor or blood, are still in early phases of translational investigation. Yet, there is increasing interest in unravelling the complex TME network regulating tumor vascularization, in search for new therapeutic targets for immunotherapy-based combinations with the aim to overcome primary or acquired resistance to ICIs.

An increasing body of preclinical and indirect clinical evidence has suggested that the proangiogenic soluble factors vascular endothelial growth factor (VEGF) and angiotensin 2 (ANG2) act as underlying shared regulators of angiogenesis and immune suppression [90,91].

Findings on the role of VEGF in neo-angiogenesis have made it a key therapeutic target for anti-cancer strategies and has led to the design of monoclonal antibodies or multi-kinase inhibitors targeting VEGF-alpha or its cognate endothelial receptor tyrosine kinases, VEGFR1, 2, and 3. The underlying rationale for anti-cancer therapy envisions tumor starvation for the shortage of blood supply as a consequence of the inhibition of vessel formation [90].

In NSCLC treatment, bevacizumab, the most widely prescribed anti-angiogenic drug, in combination with carboplatin/paclitaxel chemotherapy has long been one of the front-line standards for non-squamous histology. Upon the failure of first-line chemotherapy, two anti-angiogenic agents, nintedanib and ramucirumab, are licensed for use in combination with docetaxel despite marginal survival/PFS benefit [6,23].

The discovery of the interconnected immune modulation raised renewed interest in anti-angiogenetic agents for combination with ICIs. Recently, bevacizumab has been approved by both the FDA and EMA in association with first-line chemo-immunotherapy for non-squamous NSCLC, based on the results of the Impower150 trial [92].

VEGF is generated under oxygen shortage, as in case of uncontrolled tumor proliferation, through HIF-1 alpha signaling. Cancer cells, but also various immune-suppressive cells such as tumor-associated macrophages, myeloid-derived suppressor cells (MDSCs), monocytes, and immature dendritic cells, have been reported to produce VEGF [93].

Numerous preclinical works have shown that VEGF exerts its immune suppressive functions on different compartments. As for CD8^+^ T cells, increased VEGF can inhibit trafficking, proliferation, and effector function [94]. While inhibiting the maturation and antigen presentation of dendritic cells [95], VEGF also induces the recruitment and proliferation of immunosuppressive cells, including T regulatory cells, MDSCs, and pro- tumor associated macrophages [91]. More importantly, the aberrant tumor vasculature fostered by VEGF leads to hypoxia and to low pH, significantly hampering T cell infiltration and survival in TME [96].

Indirect evidence on the immune functions of VEGF also derives from clinical trials testing anti-angiogenetic drugs. Indeed, in renal cell carcinoma, the combination of atezolizumab and bevacizumab enhanced T cell migration, and further evidence reported increased T cell infiltration into solid tumors, possibly due to a normalized microvasculature and the restoration of homing molecules [97,98].

Notably, a study in patients with metastatic melanoma treated with immune-checkpoint inhibitors revealed that high levels of circulating ANG2 correlate with resistance to ICIs and worse prognosis, possibly reflecting hypoxia and immunosuppression in the TME [99].

Important open questions remain to be addressed in future studies, including the optimal anti-angiogenetic dose for vasculature normalization over disruption with the subsequent improvement of oxygen levels, drug delivery, and CD8^+^ T cell infiltration. Furthermore, the most appropriate neo-angiogenesis biomarkers, and especially the most accurate techniques/assays to study these phenomena, still remain to be determined.

### 2.3. Host-Related Biomarkers

#### 2.3.1. Circulating Lymphocyte Population

The limited availability of biopsy tissue in NSCLC has promoted the investigation and development of blood-based biomarkers for ICIs response. Besides the assessment of TMB, other promising circulating biomarkers are related to the host immune response, especially in terms of lymphocyte populations and proliferation.

Kamphorst et al. [100] showed that the increased proliferation of PD-1^+^ CD8^+^ T cells in peripheral blood after the first or second immunotherapy cycle was associated with clinical benefit in terms of improved objective responses. Notably, an increased proliferation was observed in about 70% of the 29 patients who were evaluated, where PD-1^+^ CD8^+^ T cells displayed an effector-like phenotype (HLA-DR^+^, CD38^+^, Bcl-2^low^) and expressed costimulatory molecules (CD28, CD27, ICOS).

Another study in epithelial thymic carcinoma and NSCLC patients by Kim et al. [101] reported that a proliferative response of peripheral blood PD-1^+^CD8^+^ T cells, measured as the fold-change in the percentage of Ki-67^+^ cells, as early as one week after the start of ICI treatment, can predict clinical benefit in terms of PFS and OS.

Another perspective has been offered by Ferrara et al. [102], who showed that an immune senescent population of circulating CD8^+^ T cells, characterized by low proliferation, CD28^−^, CD57^+^, and KLRG1^+^, independently predicted poor outcome to ICIs in terms of lower objective responses and shorter PFS and OS. Lymphocyte phenotyping was performed at baseline, and immune senescent CD8^+^ T cells were found in up to 28% of the ICI cohort. The immune senescent phenotype did not play a prognostic or predictive role for response in the platinum-based chemotherapy control, where it was present in 11% of the cohort.

In this context, the underlying limitation of this type of study is the hurdle of tracking tumor antigen-specific T cells, not only because of technical challenges but especially because of the highly heterogeneous and dynamic nature of cancer cells and of the related neo-antigens.

Clearly, the use of the (circulating) T cell phenotype as a biomarker is interesting because of the close connection with T cell and ICIs biology; nevertheless, it is at early phases of development, and perspective confirmation is needed.

#### 2.3.2. Innate Immune Populations

The negative prognostic and predictive role of circulating neutrophils and of the neutrophil/lymphocyte ratio in particular after ICIs, has been addressed in numerous retrospective studies, reporting a significant association. While the neutrophil/lymphocyte ratio or derived neutrophil/lymphocyte ratio are easy to assess from routine blood cell count and are reproducible, a clinically meaningful threshold is awaited upon for prospective validation [103,104].

Using multi-parametric flow cytometry, a recent study in NSCLC patients investigated, along with neutrophil/lymphocyte ratio, the possible association between the ICI response of HLA-DR monocytes and dendritic cells. The baseline elevated neutrophil/lymphocyte ratio and the high frequency of HLA-DR monocytes and/or the low frequency of the dendritic cells of total leukocytes identified poor responders to ICIs. Interestingly, after ICIs, reduced neutrophil/lymphocyte ratio and HLA-DR monocytes frequency as well as increased dendritic cells predicted responses and longer OS [105].

Further proof of the association between neutrophil infiltration and reduced clinical benefit after ICI treatment has been provided, measuring circulating interleukin-8, a powerful chemoattractant for neutrophils and other potentially immune-suppressive myeloid leukocytes [106]. A large retrospective analysis across various cancer types, including NSCLC, reported that elevated baseline serum IL-8 levels had a negative prognostic and predictive role on ICI response [107]. This cytokine is currently under investigation as a potential therapeutic target in combination with ICI (NCT03400332).

New high-throughput labelling technologies, such as mass cytometry, have widened our horizon on the complex innate immune cell network that orchestrates the innate immune system and cancer interaction, paving the way for future biomarker discovery.

Even if dedicated studies have not been conducted specifically in NSCLC, valuable evidence from metastatic melanoma suggests a positive correlation between the frequency of CD14^+^CD16^−^HLA-DR^hi^ monocytes in peripheral blood mononuclear cells and PFS and OS upon ICIs [108].

MDSCs represent a heterogeneous population of immature myeloid cells that fail to differentiate into neutrophils (polymorphonuclear MDSC), macrophages, or dendritic cells (monocytic MDSC) and increase during cancer and chronic infection, inhibiting T cells and natural killers (NK), among others [109,110]. Their negative predictive role for ICI outcome has mostly been proven in melanoma [111,112]. However, in a study by Youn et al. [113], a decreased frequency of circulating Lox-1 polymorphonuclear MDSC following the first cycle of nivolumab predicted a better outcome in NSCLC. The frequency of NK showed an inverse pattern, where a higher ratio of NK/Lox-1 polymorphonuclear MDSC was strongly correlated with response, PFS, and OS.

#### 2.3.3. Intestinal Microbiota/Microbiome Composition

Seminal studies have recently highlighted how the composition of the intestinal commensal bacterial flora (microbiota) plays a pivotal role in tuning systemic immune functions, including cancer immunity and response to ICIs in particular [114,115].

In the setting of metastatic melanoma, the analysis of fecal samples at baseline has revealed that the presence of specific gut bacteria (namely *Faecalibacterium genus* for ipilimumab; *Bifidobacterium longum*, *Collinsella aerofaciens*, and *Enterococcus faecium* for anti-PD-1 agents) was significantly associated with ICI benefit [116,117].

In other epithelial tumors, including NSCLC, the relative abundance of *Akkermansia muciniphila* identified ICI responders [118]. Additionally, in this study, a broader and more diverse microbiome composition was also associated with longer ICI benefit. To corroborate these findings, fecal transplantation in “germ-free” mice from ICI responders or those enriched with *Akkermansia muciniphila*, conferred increased sensitivity to ICI activity. Of note, this study also showed an association between antibiotic use and worse ICI outcome at the end of therapy [118].

It is unclear whether underlying factors might have interfered with the findings that were observed, and how to apply these results in clinical practice also remains unclear. The connection between the microbial ecosystem and anticancer immunosurveillance is unquestioned; nonetheless, inherent mechanistics may warrant further investigation.

#### 2.3.4. Germline Genetics

Since the ability to recognize and combat non-self-pathogens largely depends on the Human Leucocyte Antigen (HLA) system, increasing interest has been raised by its possible role in influencing ICI response, especially for the class I compartment.

Each HLA-I molecule binds specific intracellular peptides for presentation on the cell surface to CD8^+^ T cells. The corresponding genes are highly polymorphic, where each variant binds a select repertoire of ligands. Homozygosity in at least one HLA class I locus would be expected to present a smaller, less diverse repertoire of tumor-derived neoantigens to CD8^+^ T cell when compared to heterozygosity at each class I locus [119].

In a landmark study from 2018, Chowell et al. [120] determined the HLA class I genotype of 1535 advanced cancer patients treated with ICIs, including NSCLC, showing that HLA class I homozygosity in at least one locus was significantly associated with reduced OS, independently from mutation load, tumor stage, age, and ICI class. Notably, the predictive ability of HLA zygosity was enhanced when coupled with TMB count. Moreover, this study also described a specific HLA “supertype” associated with ICI outcome in melanoma, where patients with *HLAB44* alleles was associated with longer OS, and patients with *HLAB62* were associated with a reduced one.

The large meta-analysis by Litchfield et al. [57] did not confirm a significant association with ICI response for the germline HLA-I evolutionary divergence level, nor for the maximal HLA heterozygosity, nor for the HLA B62 supertype. The HLA B44 supertype was found to be marginally non-significant (OR = 1.17 (1.00–1.37), *p* = 0.053). However, while not significant overall, the HLA B44 supertype and germline HLA-I evolutionary divergence were significant in the anti-CTLA-4 melanoma cohorts. The same authors suggest that this latter association is potentially consistent with the increase in T cell receptor (TCR) diversity observed in anti-CTLA-4-treated patients.

Whether further investigation is warranted to clarify the relationship between HLA class I status and ICI outcome, recent evidence from melanoma—where single nucleotide polymorphisms in the FcγR impacted response to ipilimumab [121]—suggests that polymorphisms in other genes involved in immune response (i.e., HLA class II or genes involved in the antigen presentation and processing machinery) may also be implied in response to ICIs. Dedicated studies should be conducted to explore these possible host-related predictive biomarkers.

## 3. In Vivo Biomarkers for Immunotherapy: Molecular Imaging

Nowadays, noninvasive imaging techniques are playing an increasing role in screening, prognosis, baseline assessment, staging, restaging, response to treatment, and follow-up of several diseases, especially oncological ones. In this context, positron emission tomography (PET) is gaining a central role for several reasons, emerging as a predominant imaging modality. Technological advancement has allowed the development of new hybrid tomographs, in which PET is associated with the state of the art of computed tomography (CT- PET/CT) or magnetic resonance (MR- PET/MR) technologies, which are able to provide increasingly detailed information both on the functional (PET imaging) and on the morphological side (CT and MR imaging) [122,123].

In NSCLC, both anatomic evaluation with contrast-enhanced (ce)-CT and metabolic evaluation with 2-deoxy-2-[18F]fluoro-D-glucose ([18F]FDG) PET/CT are essential for patient staging and management [124,125,126]. [18F]FDG has already reached a pivotal role in NSCLC, being used routinely to evaluate tumor metabolism in NSCLC patients at staging [127,128,129]. Although guidelines do not support the use of serial [18F]FDG PET/CT in NSCLC patients treated with ICIs in clinical practice, several studies are highlighting the potential role of metabolic information in both predictive (at baseline PET) and prognostic (at interim evaluation or time to radiological progression) perspectives.

Moreover, the development of new molecular imaging probes to answer specific biological questions has further enhanced the potential role of functional imaging to predict and monitor response to immunotherapy [130].

Below, we present an overview of the role of [18F]FDG and other functional imaging probes (in vivo tools) that are currently under investigation in clinical trials or that have already been approved to enhance clinician decision-making capabilities for immunotherapeutic regimens in the NSCLC setting, highlighting limitations and advantages.

### 3.1. 2-Deoxy-2-[18F]fluoro-D-glucose ([18F]FDG) and Immunotherapy in the NSCLC Setting

#### 3.1.1. [18F]FDG and Immunotherapy Response Assessment

The NSCLC setting represents one of the most widely acknowledged indications for [18F]FDG PET/CT, which is considered to be particularly helpful for the detection of lymph node metastases and distant metastases, preventing unnecessary surgeries and improving cost-effectiveness in the management of these patients [131].

[18F]FDG PET/CT is recommended for staging NSCLC due to its excellent ability (sensitivity 93%, specificity 96%) to detect adrenal and bone metastases if the baseline CT is negative (level of evidence A). [18F]FDG PET/CT is also proposed in oligometastatic NSCLC patients who are potentially eligible for local treatment [124,125,126]. In the latest revision of the National Comprehensive Cancer Network (NCCN), [18F]FDG PET/CT is recommended for staging all patients with NSCLC, if not previously performed and in case of recurrence after therapy, but it is not routinely indicated for surveillance after therapy [132].

Despite that, [18F]FDG PET/CT has reached a pivotal role in the evaluation of response to immunotherapy in clinical routine although the method has inherent limitations in this precise setting. The growth, survival, and proliferation of cancer cells depend on their peculiar metabolism: cancer cells, even in presence of ample oxygen, prefer to metabolize glucose and ferment glucose to lactate through “aerobic glycolysis” due to mitochondrial dysfunction. This phenomenon, known as the Warburg Effect [133], is the mechanism underlying [18F]FDG uptake in tumor cells, such as those in non-small-cell lung cancer. However, immunotherapy results in an increased presence of tumor-infiltrating lymphocytes (TILs) and of the inflammatory response, which lead to a transient increase in tumor volume and increased [18F]FDG uptake in responding tumoral tissue. This phenomenon determines an underestimation of the response to therapy and can be confused with a progression, which is called “pseudo-progression” [134,135,136].

Both the morphological Response Evaluation Criteria in Solid Tumors (RECIST) 1.1 and the functional PET Response Criteria in Solid Tumors version (PERCIST) 1.0 cannot address pseudo-progression adequately [137,138,139]. Regarding functional imaging, in the evaluation criteria, the visual interpretation (presence of disease, affected districts, target lesions, and identification of new lesions) is supported and integrated by semi-quantitative interpretation, which is performed using a standardized uptake value (SUV). The SUV is calculated by normalizing the attenuation corrected FDG uptake in a lesion to the injected dose and body weight [140]. PERCIST 1.0 defines an increase of at least 30% in the sum of the SULpeak of all target lesions detected at baseline and/or the appearance of new lesions as a metabolic progression disease (MPD). However, as previously mentioned, in the pseudo-progression phenomenon, inflammatory changes could contribute to lesion enlargement. Therefore, in the last decade, the immune-related response criteria (irRC) and the immunotherapy-modified Response Evaluation Criteria in Solid Tumors (imRECIST) have been developed for the morphological response assessment to immunotherapy. Furthermore also new functional criteria have been developed to overcome this limit, namely the immunotherapy-modified PET Response Criteria in Solid Tumors (imPERCIST) and the PET Response Evaluation Criteria for Immunotherapy (PERCIMT) [129,141,142,143,144]. In particular, the imPERCIST incorporated two new patterns of response: unconfirmed metabolic progression disease (UMPD), which is a possible MPD that needs to be confirmed after follow-up with interval studies, and confirmed metabolic progression disease (CMPD), which is a real progression confirmed at the follow-up study after 4–8 weeks. At the follow-up evaluation, three different patterns can be found: the UMPD is confirmed if the uptake status is not changed; a decrease >30% of the uptake defines a partial metabolic response (PMR), confirming that the UMPD was a pseudo-progression; an increase or decrease in tumor burden different from the first two cases is defined as a stable disease, and durable stable disease may represent antitumoral activity. Finally, the PERCIMT criteria tried to overcome the limits related to the appearance of new lesions by considering a CMPD as the appearance of ≥4 new lesions <1 cm in functional diameter, ≥3 new lesions >1 cm in functional diameter, or ≥2 new lesions > 1.5 cm in functional diameter, measured in centimeters on the fused PET/CT images. The major radiological and functional response criteria to immunotherapy are summarized in Table 2.

In a recent systematic review and metanalysis by Park et al. [147], the pooled incidence of pseudo-progression was calculated to be around 5.0% (95% CI: 3.4%, 6.7%) in patients with NSCLC, around 6.4% (95% CI: 4.6%, 8.3%) in patients with melanoma, around 7.0% (95% CI: 5.2%, 8.6%) in patients with genitourinary cancer, and around 2% (one of 45; 95% CI: 0%, 6%) in patients with squamous cell cancer of the head and neck. Interestingly, they also found that the pooled incidence of pseudo-progression was higher in melanoma patients treated with CTLA-4 inhibitor (9.7%) compared to studies of PD-1/PD-L1 inhibitor monotherapy (5.7%; 95% CI: 4.8%, 6.6%); in particular, in studies of NSLSC patients treated with PD-1 inhibitor, the incidence of pseudo-progression ranged from 3.4% to 6.9%. Indeed, Humbert et al. [148] recently reported that pseudo-progression occurs in more than one third of the NSCLC patients treated with Nivolumab or Pembrolizumab: 58% (29/50) showed a real PD, but more than one-third (11/29) were misclassified, as they finally reached a durable clinical benefit (DCB). No standard interim PTE criteria has been found to accurately distinguish responding from non-responding patients.

Moreover, these criteria can help to manage the phenomenon of pseudo-progression. However, no criteria are currently able to define “hyperprogression”, a response pattern that is mainly associated with the use of anti-PD-1/PD-L1 agents as well as with the use of anti-CTLA-4 therapy. Hyperprogression is defined as a two-fold or greater increase of the tumor volume growth rate during immunotherapy, and it is associated with a worse clinical outcome, high mortality, and a median overall survival of 3–6 months. Some hypotheses have been developed, such as T regulatory cell expansion and T cell exhaustion; however, the prevalence, the natural history, and the predictive factors of hyperprogression in patients with cancer treated by anti-PD-1/PD-L1 remain unknown [127].

In 2017, Champiat et al. [149] analyzed the medical records from all patients (N = 218) prospectively treated at Gustave Roussy with anti-PD-1/PD-L1 within phase I clinical trials. Hyperprogression was found in 9% of the population eligible for the study (9/131 patients), which was associated with a higher age (*p* < 0.05) and a worse outcome, but no association was found with higher tumor burden at baseline nor with any specific tumor sub-type. More recently, other quantitative PET parameters such as metabolic tumor volume (MTV) and total lesion glycolysis (TLG), which is calculated by multiplying SUVmean to the MTV of the selected region of interest (ROI), have also been used to measure tumor burden, defined as the total volume of tumor lesions with increased metabolic activity, and their use has been recently implemented in both scientific papers and clinical practice [140]. In 2020, Castello et al. [150] investigated the prevalence of hyperprogression phenomenon and its association with clinical variables and metabolic parameters by [18F]FDG PET/CT in 50 patients with NSCLC treated with ICIs, which were prospectively collected. Hyperprogression was found in 14/50 patients and, in contrast to the study of Champiat et al., a statistically significant association was found between hyperprogression and tumor burden, expressed by both TLG (756.1 cm^3^ for HPD vs. 475.6 cm^3^ for non-HPD, *p* = 0.042) and MTV (287.3 for HPD vs. 62.1 for non-HPD, *p* = 0.011). The median OS for patients with hyperprogression was 4 months.

#### 3.1.2. [18F]FDG and Immune-Related Adverse Events (irAEs)

As previously mentioned, consistent with the ICIs mechanism of action, the immune system activation responsible for the anti-tumor response may also be involved in the onset of autoimmune reactions, where several healthy tissues can be invaded by activated by the resulting T cells. IrAEs commonly manifest within the first 3 months of ICI administration; however, they could appear at any time and are most frequently associated with combination treatments (anti-CTLA-4 plus anti-PD-1 or anti-PD-L1) followed by anti-CTLA-4 drugs and finally by anti-PD-1 or anti-PD- L1 drugs. The onset of these immune-related adverse events (irAEs), such as hypophysitis, pneumonia, colitis, hepatitis, and thyroiditis, is unpredictable and, though mild in most cases, can still be life-threatening, thus requiring prompt recognition and appropriate immune-suppressive therapy.

Compared with morphologic imaging, [18F]FDG PET/CT images can early detect irAEs due to the increased uptake of [18F]FDG at these sites. This potentially allows for a rapid intervention in life-threatening cases, such as severe colitis, pneumonitis, and pancreatitis [127,151].

In 2018, Aide et al. [152] have compiled a checklist for researching immune-related side effects: measure the spleen and the liver-to-spleen [18F]FDG uptake ratio, an inversion of this ratio reflecting immune activation preceding T cell proliferation; consider whether the pattern of new nodal uptake suggests sarcoidosis (lambda sign); refer to baseline scan when an organ frequently showing increased physiological uptake is thought to be involved by an immune-related side effect (thyroid, stomach); check the pituitary gland; consider that irAEs, such as colitis and pneumonitis, may need treatment withdrawal or corticosteroid treatment; consider that bilateral adrenal enlargement and increased uptake is probably due to adrenalitis.

In NSCLC patients, the most common irAEs are rash and diarrhea with nivolumab treatment and hypo- and hyperthyroidism with pembrolizumab treatment [153]. In particular, thyroiditis has an incidence of 5–23%, and [18F]FDG uptake in the thyroid gland can depict thyroid inflammation even before laboratory changes are observed and allows the prediction of thyroid dysfunction development, as demonstrated by the study by Eshgh et al. [154], which was performed on 18 patients with NSCL and who were treated with nivolumab.

Interestingly, it has been recently shown that the onset of irAEs may also predict the clinic benefit of ICIs [155,156,157]. Even though the clinical implications of this evidence are limited at present, in the future, the identification of predictive factors for the onset and especially for the severity of irAE may further guide treatment decisions on how ICI should be administered to maximize response and safety together.

### 3.2. [18F]FDG, Radiomic and AI

The utility of noninvasive imaging has been further enhanced by the introduction of radiomics and artificial intelligence (AI), which has improved the interpretation of medical image datasets. Radiomics is a new innovative bioinformatic approach for image analysis that allows tumor heterogeneity to be evaluated and parameters to be quantified, namely radiomic features (RFs), using standardized mathematical based models. RFs include first-order statistical functions, such as conventional parameters (i.e., the standard uptake value (SUV), the metabolic tumor volume (MTV), or the total lesion glycolysis (TLG)), histogram and shape PET parameters, and second-order and high-order statistical functions, which also contain information about the spatial relationships between the intensities of more than two voxels (such as texture parameters). RFs can be applied both in clinical practice (especially conventional parameters) and in clinical research, supporting and implementing the clinical decision support system (CDSS) [158,159,160]. More recently, AI-based approaches have been applied to medical imaging in combination with radiomics or alone, allowing relevant features to be identified with several fully automatic or semi-automatic approaches (machine learning and deep learning) [161,162,163,164]. These new approaches could implement the assessment of different aspects and stages of the lung cancer history: to differentiate the histological subtypes of cancer, particularly, adenocarcinoma and squamous cell carcinoma; to predict EGFR mutation status or PD-L1 expression for the risk stratification of NSCLC patients; and to predict response to different kind of therapies [165].

#### 3.2.1. Molecular Profiling

Radiogenomics is one of the most interesting fields of application of radiomics, allowing the correlation of quantitative imaging features (representative of the phenotype of the genomic signature) with genomic profiles. Thus, radiogenomics represent a non-invasive, whole-body, and repeatable method for extracting molecular information from images. Indeed, several studies have tried to identify the radiomic signature of NSCLC patients trough [18F]FDG PET/CT images, especially regarding EGFR mutation.

As mentioned above, actionable mutations in the driver oncogenes *EGFR* and *ALK* confer resistance to ICIs [60,61]. For that reason, studies on [18F]FDG PET/CT images that have aimed to identify the clinico-radiologic predictors of tumors with *KRAS*, *EGFR*, *BRAF*, *HER2*, *MET*, *ALK*, *ROS1*, and *RET* mutations in NSCLC patients are not only useful with regard to targeted therapies but also for a possible predictive value for the response to ICI [166,167,168,169,170,171,172,173].

While several studies have demonstrated how imaging phenotypes are connected to somatic mutations through an integrated analysis of NSCLC patients with somatic mutation testing and [18F]FDG PET/CT images using both radiomic and AI, only one study by Jiang at al. [174] explored the potential value of radiomic features from [18F]FDG PET/CT images in assessing different PD-L1 mutational status in 399 NSCLC. Models that comprised radiomic features from the CT, the PET, and the PET/CT images resulted in an AUC of 0.97, 0.61, and 0.97, respectively, for prediction of PD-L1 (SP142) expression level over 1% and in an AUC of 0.91, 0.75, and 0.88, respectively, for PD-L1 expression level over 50%.

#### 3.2.2. Staging and Prognostic Risk Stratification: Prognostic and Predictive Value

Several studies have evaluated the prognostic and predictive role of conventional PET parameters (SUVmax or SUVmean) in NSCLC patients who are ICI candidates.

In 2016, Lopci et al. [175] investigated the correlation between baseline (prior to surgery) [18F]FDG PET/CT parameters and IC tissue expression and other markers of tumor-related immunity in 55 resected NSCLC patients (stage IA-IIIB: 36 adenocarcinomas, 18 squamous cell carcinomas, and 1 sarcomatoid carcinoma). Sampled surgical tumor specimens were analyzed by immunohistochemistry (IHC) for CD68-TAMs (tumor-associated macrophages), CD8-TILs (tumor infiltrating lymphocytes), PD-1-TILs, and PD-L1 tumor expression. The median SUVmax 11.3 (range: 2.3–32.5) resulted in significantly higher squamous cell carcinomas compared to other NSCLC histotypes (*p* = 0.007). Moreover, a statistically significant correlation between SUVmax and SUVmean with the expression of CD8^+^ TILs (rho = 0.31; *p* = 0.027) and PD-1 (rho = 0.33; *p* = 0.017 and rho = 0.36; *p* = 0.009, respectively) was found.

In 2019, Takada et al. [176] aimed to study the association between [18F]FDG PET/CT and the response to anti-PD-1 therapy in 89 NSCLC patients with advanced or recurrent disease. Among the 89 patients evaluated, 24 patients were classified as responders with an average SUVmax of 15.60, and 65 patients were classified as non-responders with an average SUVmax of 11.61 (*p* = 0.0016). The response rate in patients with SUVmax ≥ 11.16 (41.3% [19/46]) was significantly higher compared to patients with SUVmax < 11.16 (11.6% [5/43], *p* = 0.0012).

After these studies, Grizzi et al. [177] reported their preliminary data analysis on 27 NSCLC patients, stage IV or IIIB and ineligible for local therapy, treated with monoclonal IgG targeting PD-1 or PD-L1 as single agent (54 patients with nivolumab, 21 with pembrolizumab, and 5 with atezolizumab, respectively). In contrast with the aforementioned study, their data documented how classifying patients as “fast progressors” or “responders” after 8 weeks of treatment in almost all cases with fast progression SUVmax was ≤17.1 (sensitivity 88.9%) or SUVmean ≤ 8.3 (sensitivity 100%).

More recently, new semiquantitative parameters, namely metabolic tumor volume (MTV) and total lesion glycolysis (TLG) rate, have been studied to evaluate the predictive and prognostic value of the metabolic tumor burden at baseline [18F]FDG PET/CT. Many of these studies have shown that new PET metabolic parameters are better in predicting prognosis in early and advanced NSCLC and SCLC patients than conventional PET parameters (SUVmax or SUVmean) [178,179].

Higher MTV was significantly related with worse PFS (*p* < 0.001) and OS (*p* < 0.001) in 61 patients with treated NSCLC [180]. The same group [181] also found that a 25-mL increase in MTV in the restaging PET of 19 patients with lung cancer (18 NSCLC and 1 SCLC) was associated with a statistically increased risk of progression (5.4-fold; *p*= 0.0014) and death (7.6-fold; *p* = 0.001), respectively.

In 2020, Polverari et al. [182] evaluated predictive factors of response to immunotherapy in 57 advanced NCSLC patients, demonstrating that patients with higher MTV and TLG had a higher probability of disease progression compared to those patients presenting with lower values. SUVmax did not show a correlation with PD status, PFS, and OS. Moreover, patients with high MTV, TLG, and heterogeneity evaluated by the radiomic features “skewness” and “kurtosis” had a higher probability of ICIs failing.

Moreover, radiomic and AI have been successfully used also for prognostic risk stratification. In a study performed by Kim et al. [183], [18F]FDG PET/CT parameters, clinical, mRNA-seq, and whole exome-seq data of 11 adenocarcinoma and 20 squamous cell carcinoma from The Cancer Genome Atlas (TCGA) database were analyzed. In linear regression analysis, texture parameters such as low gray-level run emphasis (LGRE, R2 = 0.48, *p* < 0.0001), short-run low gray-level emphasis (SRLGE, R2 = 0.45, *p* < 0.0001), and long-run low gray-level emphasis (LRLGE, R2 = 0.41, *p* = 0.0001) showed a remarkable correlation with PD-L1 mRNA expression.

More recently, Valentinuzzi et al. [184] created a radiomic signature (iRADIOMICS) able to predict response of metastatic NSCLC (stage IV) to pembrolizumab compared to the clinical standards (iRECIST and PD-L1 immunohistochemistry). A total of thirty patients receiving pembrolizumab were scanned with [18F]FDG PET/CT at baseline, month 1, and month 4. Multivariate baseline iRADIOMICS was to be found superior to the current standards in terms of predictive power, in terms of both time (AUC (95% CI) and accuracy = 0.90 (0.78–1.00), 78% (18%)).

Other parameters, such as the spleen-to-liver ratio (SLR) or the bone marrow-to-liver (BLR) SUVmax ratio, were shown to represent surrogates (indirect index) of the hematopoietic tissue metabolism. Seban et al. [185] proposed a prognostic score combining baseline total metabolic tumor volume (TMTV) and the Derived Neutrophil-to-Lymphocyte Ratio (dLNR) to predict OS in 80 NSCLC patients receiving ICIs. The median follow-up was 11.6 months (95%CI 7.7–15.5). Progression and death occurred in 64 and 52 patients, respectively; disease clinical benefit (DCB) was reported in 40% of cases. Baseline tumor burden (TMTV) > 75 cm^3^ on [18F]FDG PET/CT scans and inflammatory status (dNLR) > 3 were associated with poor OS and the absence of DCB for ICI treatment in advanced NSCLC patients, unlike tumor SUVmax, and might be used in the future in order to improve the selection of appropriate candidates.

In 2021, Bauckneht et al. [186] aimed to identify reliable biomarkers (systemic inflammation indexes and PET imaging parameters at the time of radiological progression) able to discriminate between responders and non-responders among 45 patients showing imaging progression (defined as RECIST 1.1 progressive disease) during nivolumab administration for advanced NSCLC. Among all of the parameters collected, MTV and the systemic inflammation index (SII) independently predicted OS. Moreover, they developed the immune-metabolic prognostic index (IMPI) based on these two parameters, categorizing the enrolled cases in three groups with different risks of progression as follows: low (neither MTV ≥ 208.01 nor SII ≥ 197.21, IMPI = 0, *n* = 11), intermediate (MTV ≥ 208.01 or SII ≥ 197.21, IMPI = 1, *n* = 23), and high IMPI (MTV ≥ 208.01 and SII ≥ 197.21, IMPI = 2, *n* = 11). The median OS was 17.5 months (95% CI 11.3–31.5 months), 9.4 months (95% CI 5.6–33.6 months), and 3.2 months (95% CI 2.1–18.5 months) for the low, intermediate, and high IMPI groups, respectively (*p* < 0.0001). This interesting index, if validated, may allow the identification of patients who might benefit from immunotherapy continuation, regardless of radiological progression.

### 3.3. Functional Imaging Probe beyond [18F]FDG

[18F]FDG PET/CT is still the mainstay in the evaluation of NSCLC patients treated with ICI, and new technologies, such as radiomics and AI, have expanded the ability of this radiotracer to address open questions about ICI mechanisms. Nevertheless, [18F]FDG is not able to assess the finer mechanisms underlying tumor IC expression and their inhibition.

For this reason, in the last few decades, new molecular imaging probes have been developed to improve our knowledge of TME, the immune-system, and ICIs. Even if biopsy and tissue analysis are still mandatory and strictly necessary, the in vivo molecular imaging of IC expression may be more accurate than in vitro analysis. Moreover, inducible T cell costimulatory receptor (ICOS) PET radiotracers may predict early response to therapy.

As previously mentioned, a biopsy specimen is the first fundamental step to identify PD-L1 tumor expression and/or PD-1 and CTLA-4 T cell expression and for therapy and patient selection. However, in clinical practice, generally only a single-tumor biopsy is performed, leading to the potential underestimation of the tumor heterogeneity of ICs expression, both spatial- (inter- and intra-tumoral) and time-related (more aggressive cell clones developing over time) [187,188].

As previously mentioned, the spatial immune infiltration patterns (‘topography’) across cancer entities and across various immune cell types is a central concept to understand the mechanisms underlying the response to ICIs immunotherapy [189].

Three distinct “immune topographies” have been identified: hot tumors characterized by lymphocyte infiltration mixed with tumor cells in the tumor core; cold tumors, characterized by an absence of lymphocyte infiltrations (i.e., almost no lymphocytes can be seen on histological slides); and immune-excluded tumors, characterized by an abundance of lymphocytes at the invasive edge of the tumor but few to no lymphocytes in the tumor core [190].

ImmunoPET radiotracers, ICOS PET radiotracers, and [18F]FDG give the possibility to systematically map the “Immune Topographies” in both the spatial (whole-body) and temporal distribution in NCSLC [130]. These techniques trough whole-body specific target expression probes could improve the current knowledge on TME and all of the dynamic mechanisms behind tumor development and growth, including the relationship between tumor cells and infiltrating immune cells in the TME and the grading heterogeneity among primary and metastasis.

If validated, in the future, these new radiotracers could help clinician decisions in clinical practice, stratifying the patients who are good candidates for immunotherapy at baseline, identifying earlier response assessment, distinguishing tumor progression from pseudo-progression (especially immunoPET radiotracers combined with [18F]FDG PET/CT images), monitoring “immune topographies” changes during ICIs, and identifying potential non-responding patients (patients which ICOS PET shows low or no uptake of activate T cells) and/or lesions.

New radiotracers that may have a role in the evaluation of NSCLC patients who are candidates for immunotherapy and that are currently under study in clinical trials are described below.

#### 3.3.1. PD-1/PD-L1 Pathways

Several techniques have been implemented to find the best radiotracer, specifically for an antigen expression only on target cells labelled with an optimal radionuclide for imaging. For example, antibody fragments and small proteins labeled with [18F] and [68Ga] (half-life of 109.8 and 67.7 min, respectively) are preferred over immunoglobulin G (IgG) antibodies (mAbs) labeled with [111In], [89Zr], and [64Cu] (half-life of 67.2, 78.4, and 12.7 h, respectively) due to their shorter biological half-life [13,191,192].

Since 2015, after the development of the first [64Cu]DOTA labeled antibody (IgG) PD-1 radiotracer [193], several immune checkpoint radiotracers have been developed for PET imaging, but only a few of them have been tested in humans.

The first in-human study was presented by Bensch at al. [194] in 2019. They studied [89Zr]atezolizumab PET/CT images in 22 patients with different cancers (metastatic bladder cancer, NSCLC, or triple-negative breast cancer). To prevent rapid clearance during imaging, which was performed on days 4 and 7, 10 mg of unlabeled atezolizumab was administered together with the radiotracer. [89Zr]atezolizumab uptake was found to be physiologically high in the bone marrow, intestines, kidney, and liver and low in healthy brain, subcutaneous tissue, muscle, compact bone, and lung, while the uptake in lymphoid tissue (lymph nodes and spleen mainly) depended from the activation state of the immune system. They found that pre-treatment radiotracer uptake better correlated with PFS and OS, comparing it to the conventional IHC staining of PD-L1, highlighting the limitations of a single biopsy evaluation. Moreover, tumor [89Zr]atezolizumab uptake was generally high (mean SUVmax of 10.4) but often with an high intratumoral, intertumoral, and interpatient heterogeneity distribution.

In 2018, Niemeijer et al. [195] studied 13 patients with advanced NSCLC before treatment with nivolumab using two new immuno-PET radiotracers: an anti-PD-1 radiotracer, [89Zr]nivolumab (images acquired after 7 days after injection), and an anti-PD-L1 radiotracer, [18F]labeled anti-PD-L1 fibronectin-based protein (adnectin), namely [18F]BMS-986192 (images acquired after 1 h after injection). Both radiotracers showed favorable distribution in the tumor but with a high intratumoral, intertumoral, and interpatient heterogeneity distribution. Higher uptake of both radiotracers was found in responder patients: [18F]BMS-986192 SUV peak (median 6.5 (responder) vs. 3.2 (non-responder), *p* = 0.03), and a similar association was found for [89Zr]nivolumab (median SUV peak 6.4 vs. 3.9, *p* = 0.019). Spleen uptake reflected the presence of immune cells, and both radiotracers were catabolized in the liver. The median [18F]BMS-986192 uptake was higher for lesions with ≥50% tumor PD-L1 expression evaluated by IHC than for lesions with <50% expression. Similarly, [89Zr]nivolumab uptake was higher in lesions with a higher aggregates of PD-1 positive tumor-infiltrating immune cells during the tumor biopsy. Finally, due to low central nervous system (CNS) tracer penetration, both tracers showed lower accumulation in brain metastases compared to in the extracerebral lesions.

#### 3.3.2. CD8+ T Lymphocytes

A great effort has been made in recent years to optimize ICOS PET radiotracers for the detection of CD8^+^ and CD4^+^ T cells in the TME.

The only ICOS PET radiotracer tested in human is a radiolabeled minibody against tumor-infiltrating CD8-positive (CD81) T lymphocytes, with a fast blood clearance due to its small size, which allows for early image acquisition.

In 2020, Pandit-Taskar et al. [191] assessed the safety and utility of [89Zr]IAB22M2C for targeted imaging of CD81 T cells in cancer patients (1 melanoma, 4 NSCLC, and 1 hepatocellular carcinoma). The highest uptake was observed in the spleen, followed by the bone marrow. The maximum uptake in normal lymph nodes was reached between 24–48 h. Uptake in tumor lesions was already seen 2 h after injection, but the uptake in the [89Zr]IAB22M2C-positive lesions increased until the 24 h mark (SUV from 5.85 to 22.8). No side effects were registered, and serum clearance was biexponential.

#### 3.3.3. Cancer-Associated Fibroblasts (CAFs)

As previously mentioned, the tumor microenvironment (TME) is composed by an organic and evolving complex of tumor, stromal, and tumor-infiltrating immune cells, co-evolving and growing within a protective extracellular matrix (ECM) with the help of cytokines, glycoproteins, proteoglycans, and growth factors [130].

In this context, cancer-associated fibroblasts (CAFs) play an essential role in tumor growth, the integrity of TME architecture and function, tumor progression, metastasis development, and therapeutic resistance through potent immunosuppressive activity, conferring resistance to immune-based therapies. CAFs are activated by tumor cells through the secretion of factors such as *TGF-β*, *PDGF*, *EGF*, *CTGF*, and *FGF*, or in response to damage-associated molecular patterns (DAMPs) released by damaged tissue or necrotic tumor cells [192].

Indeed, a greater understanding of how tumor stroma and TME comprehensively affect immunotherapy will provide important advances: CAF markers correlate with T cell immunosuppression, blocking the effects of both CD8^+^ T cells and NK cells, especially through the release of various chemokines and cytokines (mainly IL-6 production), leading to poor clinical outcome. It is for these reasons that CAF depletion and targeting CAF-dependent pathways may indirectly lead to malignant cell death through both immune-dependent and immune-independent mechanisms [196].

In this context, the fibroblast activation protein (FAP), a transmembrane serine protease and marker of CAF activation, is exclusively synthesized and is highly expressed by CAFs from almost all tumors, whereas it is relatively absent in normal stromal cells. FAP is an important CAF protein associated with the outcome of several immunotherapies, and its presence in histological staining correlates with poor prognosis for most cancer patients.

A study has shown that in lung cancer models where the presence of FAP is genetically depleted, this leads to hypoxic necrosis in the tumor in addition to the induction of CD8^+^ T cell infiltration [197],

Recently, researchers at the University of Heidelberg have introduced PET imaging for FAP expression in cancer, producing radiopharmaceuticals derived from quinoline peptidomimetics that bind with high affinity to FAP expressed on CAFs labeled with [68Ga] [198,199]. While [18F]FDG visualizes tumor cell glycolysis, [68Ga]-DOTA-FAPi predominantly depicts tumor stroma, which is influenced by the amount of tumor stroma and the level of FAP expression in the tumor stroma, and is extremely promising due to the excellent visualization of the tumor resulting from the high tumor-to-background ratio (TBR) [200,201].

FAPi PET can provide diagnostic, predictive, prognostic, and phenotypic information. A combined [18F]FDG and FAPi PET imaging approach to determine the relationship between tumor and stroma can therefore be highly informative [201,202]. Moreover, FAPi can serve as a theranostic: the high TBR can be exploited by labeling FAPi with alpha- or beta-emitting isotopes to generate a potentially potent therapy [203,204,205].

However, FAPI PET is still in a preliminary study phase, and all of the the advantages and limitations of this radifarmaceutical are still being understood, so it will take some time before widespread adoption can become feasible [206]. For example, radiation may induce fibrosis, limiting the usefulness of FAPI PET for decision making after therapy. Moreover, fibroblast activation could be associated with chronic inflammation, liver and lung fibrosis, and checkpoint inhibitor-associated myocarditis, reducing the specificity and limit the diagnostic value of FAPI PET, so further studies are needed [207,208,209].

#### 3.3.4. Other Probes

Many other tracers may be useful in the evaluation and monitoring of immunotherapy candidate patients outside of the immunoPET radiotracer, ICOS PET radiotracer and [18F]FDG triad.

New PET imaging tracers are currently being developed to assess the presence of hypoxia in tumor tissue, a hallmark of carcinogenesis and tumor resistance to systemic therapy. Currently, there are mainly three major radiotracer of hypoxia PET tracers:

The tracer 18F-fluoromisonidazole ([18F]F-FMISO), which enters cells via passive diffusion, where it is reduced by nitroreductase enzymes under hypoxic conditions. Images from [18F]F-FMISO PET/CT are challenging because the images must be acquired several hours after the radiotracer adminstration;

The 18F-fluoroazomycin arabinoside ([18F]F-FAZA) probes are the more hydrophilic hypoxia probes. Unfortunately, the imaging of this tracer is also challenging because the image quality (resolution of tissue-hypoxia ratios) depends on renal clearance, which could lead to measurement variability.

The 64Cu-labeled diacetyl-bis(N-methylthiosemicarbazone) analog ([64Cu]Cu-ATSM) is an alternative class of agents with low molecular weight and lipophilicity, resulting in high membrane permeability. Its use in the clinical setting is more easily handled [210,211].

These radiotracers have mainly been studied to assess tumor resistance in chemotherapy and radiotherapy settings. However, as recently reported by Pietrobon et al. [212], the exclusion of lymphocytes from the tumor nest is a common phenomenon that limits the efficiency of ICI in solid tumors (cold tumors), and one hypothesis is that hypoxia, a hallmark of most solid tumors, may be a common biological determinant of immune exclusion in solid tumors.

To the best of our knowledge, to date, no study regarding hypoxia tracers in correlation with immunotherapy has been conducted in NSCLC. However, one study of Rühle et al. [213] aimed to investigate the interaction between tumor hypoxia dynamics and the PD-1/PD-L1 axis in 49 patients with head-and-neck cancer (HNSCC) undergoing chemoradiation and its relevance for patient outcomes in a prospective trial. Patients performed [18F]F-FMISO PET/CT at weeks 0, 2, and 5 during treatment; the SUV index was defined as the ratio of maximum tumor [18F]FMISO SUV to the mean SUV in the contralateral sternocleidomastoid muscle (i.e., tumor-to-muscle ratio). The PET parameters and the PD-1/PD-L1 expression both on immune cells and on tumor cells obtained from pre-therapeutic tumor biopsies were analyzed. Interestingly, Rühle et al. found that patients with no hypoxia resolution between weeks 0 and 2, and PD-L1 expression on tumor cells (TPS of at least 1%) showed significantly worse locoregional control (LRC; HR = 3.374, *p* = 0.022) and a trend towards reduced PFS (HR = 2.752, *p* = 0.052), while the PD-L1 expression of the tumor cells did not influence the outcomes of patients with early tumor hypoxia resolution.

As previously mentioned, *VEGF* and *VEGFR* are upregulated, and tumor angiogenesis often produces blood vessels with aberrant morphology, excluding T cell migration. Due to the therapeutic development of combination of anti-angiogenetic drugs with ICIs, anti-*VEGF* or anti-*VEGFR* monoclonal antibodies labelled with diagnostic or that are therapeutic radionuclide-bound may become optimal radiotracers for PET images [214].

Bevacizumab is a IgG1 antibody directed against *VEGF-A*. The radiolabeled bevacizumab, [64Cu]bevacizumab, is the only PET radiotracer of angiogenesis that has been tested in clinical studies, with tests being performed in patients with breast cancer, neuro-endocrine tumors (NET), renal cell carcinoma, and NSCLC [215]. In 2014, Bahce et al. [216] aimed to evaluate whether the uptake of [64Cu]bevacizumab in 7 patients with advanced NSCLC tumors could be visualized and quantified prior to treatment with bevacizumab combined with carboplatin and paclitaxel. Zr-bevacizumab uptake (SUVpeak) was approximately four times higher in tumor tissues (primary tumor and metastases) than in non-tumor tissues (healthy muscle, lung, and fat) on days 4 and 7. The level of tracer uptake was associated with the improved progression free survival (PFS) and overall survival (OS). Moreover, tumor tracer uptake was blocked by adding non-labeled bevacizumab, supporting the notion of a specific binding.

Even if *VEGF* plays a prominent role in mediating T cell infiltration into tumors and in regulating their function, until now, no studies have been focused on angiogenesis PET radiotracers and immunotherapy.

## 4. Conclusions

In recent decades, impressive improvements in translational medicine has led to a better understanding of the mechanisms behind tumor development and interaction with TME setting and of the implementation of ICI-based treatment options.

The way forward to create an accurate multivariable platform to predict ICI outcome is still long. Nonetheless, the continuous clinical and scientific collaboration between oncologists, pathologists, geneticists, and nuclear medicine physicians could be the winning strategy for the development of a composite assessment of the circulating and in vivo biomarkers in tissue, refining our ability to predict response to ICIs, ultimately improving patient selection for more tailored ICI-based therapies.

## Figures and Tables

**Figure 1 cancers-13-04543-f001:**
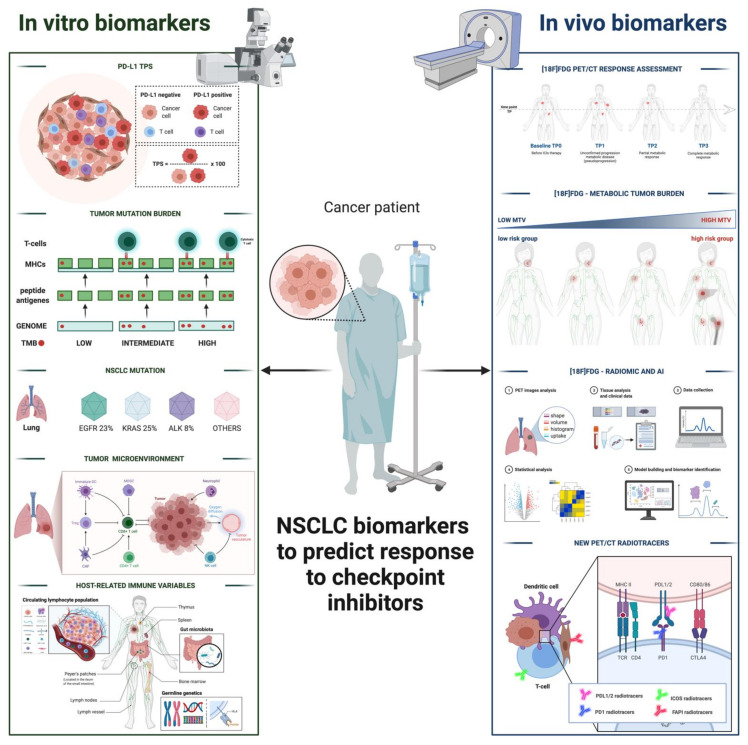
Overview of the main in vitro and in vivo biomarkers that may enrich the clinical benefit of ICI treatment NSCLC patient candidates.

**Table 1 cancers-13-04543-t001:** Summary of the FDA- and EMA-approved immune checkpoint inhibitors for non-small cell lung cancer (NSCLC) patients.

ICI Class	DRUG	Stage	Line	FDA Indication	EMA Indication
CTLA-4 inhibitor	Ipilimumab	IV	1st	In combination with nivolumab if tumor PD-L1 ≥1% (no *EGFR* or *ALK* mutation) In combination with nivolumab and 2 cycles of platinum-doublet chemotherapy (no *EGFR* or *ALK* mutation) [25]	In combination with nivolumab and 2 cycles of platinum-doublet chemotherapy (no *EGFR* or *ALK* mutation) [26]
PD-1 inhibitors	Nivolumab	IV	1st	In combination with ipilimumab if tumor PD-L1 ≥1% (no *EGFR* or *ALK* mutation) In combination with ipilimumab and 2 cycles of platinum-doublet chemotherapy (no *EGFR* or ALK mutation) [27]	In combination with ipilimumab and 2 cycles of platinum-doublet chemotherapy (no *EGFR* or *ALK* mutation) [28]
			2nd-N	As single agent if progression on or after platinum-based chemotherapy (no *EGFR* or *ALK* mutation) If *EGFR* or *ALK* mutation: as single agent if progression on corresponding FDA-approved therapy [27]	As single agent if progression on or after chemotherapy [28]
	Pembrolizumab	III *-IV	1st	In combination with pemetrexed+platinum chemotherapy in non- squamous histology (no *EGFR* or *ALK* mutation) In combination with carboplatin+nabpaclitaxel/paclitaxel in squamous histology As a single agent if tumor PD-L1 ≥1% (no *EGFR* or *ALK* mutations) [29]	In combination with pemetrexed+platinum chemotherapy in non- squamous histology (no *EGFR* or *ALK* mutation) In combination with carboplatin+nabpaclitaxel/paclitaxel in squamous histology As a single agent if tumor PD-L1 ≥50% (no *EGFR* or *ALK* mutations) [30]
			2nd-N	As a single agent if tumor PD-L1 ≥1% at progression on or after platinum-based chemotherapy If *EGFR* or *ALK* mutation: as single agent if tumor PD-L1 ≥1% and progression on corresponding FDA-approved therapy [29]	As a single agent if tumor PD-L1 ≥1% at progression after ≥1 prior chemotherapy If *EGFR* or *ALK* mutation: as single agent if tumor PD-L1 ≥1% and progression after ≥1 prior chemotherapy and target therapy [30]
	Cemiplimab	III *-IV	1st	As single agent if tumor PD-L1 ≥50% (no *EGFR*, *ROS-1*, or *ALK* mutations) [31]	As single agent if tumor PD-L1 ≥50% (no *EGFR*, *ROS-1,* or *ALK* mutations) [32]
PD-L1 inhibitors	Durvalumab	IIIA **-B	Consolidation after CH-RT	Disease not progressed following concurrent platinum-based chemo-radiotherapy [33]	Disease not progressed following platinum-based chemo-radiotherapy if tumor PD-L1 on ≥1% [34]
	Atezolizumab	IV	1st	As single agent if tumor PD-L1 ≥50% or tumor-infiltrating immune cells PD-L1 covering ≥10% of the tumor area (no *EGFR* or *ALK* mutations) In combination with bevacizumab+paclitaxel+carboplatin in non-squamous histologies (no *EGFR* or *ALK* mutations) In combination with carboplatin/nab-paclitaxel in non-squamous histologies (no *EGFR* or *ALK* mutations) [35]	As single agent if tumor PD-L1 ≥50% or tumor-infiltrating immune cells PD-L1 covering ≥10% of the tumor area (no *EGFR* or *ALK* mutations) In combination with bevacizumab+paclitaxel+carboplatin in non-squamous histologies (no *EGFR* or *ALK* mutations) In combination with carboplatin/nab-paclitaxel in non-squamous histologies (no *EGFR* or *ALK* mutations) [36]
			2nd-N	As single agent at progression during or following platinum-containing chemotherapy If *EGFR* or *ALK* mutation: as single agent or in combination, after failure of FDA-approved therapy for NSCLC harboring these aberrations [35]	As single agent at progression during or following platinum-containing chemotherapy If *EGFR* or *ALK* mutation: in combination with carboplatin bevacizumab and paclitaxel after failure of appropriate targeted therapies. As single agent at progression on chemotherapy and targeted therapy [36]

*: III stage ineligible for surgery of definitive chemo-radiotherapy, **: Unresectable. Abbreviations: CH-RT, chemo-radiotherapy.

**Table 2 cancers-13-04543-t002:** Summary of the radiological and functional immunotherapy-modified response criteria (irRC, imRECIST, imPERCIST, PERCIMT) to the immunotherapy in tumor compared with the non-immunotherapy-specific ones (RECIST 1.1 and PERCIST).

**Radiological Response Criteria**	**RECIST 1.1** [145]	**irRC** [144]	**imRECIST** [141]
Complete response (CR)	Disappearance of all target and non-target lesions without any new lesions. Any pathological lymph nodes must have reduction in short axis to <10 mm. Determined by two observations not less than 4 weeks apart.	Disappearance of all target lesions. Determined by two observations not less than 4 weeks apart.	Disappearance of all target and non-target lesions without any new lesions. Any pathological lymph nodes must have reduction in short axis to <10 mm. Determined by two observations not less than 4 weeks apart.
Partial response (PR)	At least a 30% decrease of the sum of maximum diameters of target lesions; no new lesions; no progression of disease.	Sum of product of all lesions decreased by >50% for at least 4 weeks; no new lesions; no progression of any lesions.	At least a 30% decrease of the sum of maximum diameters of target lesions; no new lesions; no progression of disease.
Stable disease (SD)	Does not meet the criteria for CR, PR, or PD, taking the smallest sum of the maximum diameters of target lesions as reference.	Sum of product of all lesions decreased by <50% or increased by <25% in the size of one or more lesions.	Does not meet the criteria for CR, PR, or PD, taking the smallest sum of maximum diameters of target lesions as reference.
Progressive disease (PD)	Sum of the maximum diameter of lesions increased by >20% over the smallest achieved sum of maximum diameter. The appearance of one or more new lesions is always considered progression.	A single lesion increased by >25% (over the smallest measurement achieved for the single lesion) or the appearance of new lesions that has to be confirmed in two consecutive observations at least 4 weeks apart.	Sum of the maximum diameter of lesions increased by >20% over the smallest achieved sum of maximum diameter. The appearance of new lesions and/or progression of non-target lesions are considered iUPD and must be confirmed 4–8 weeks later as iCPD. Progression is not confirmed in case of the shrinkage of these lesions at 4–8 weeks, and evaluation must be reset.
**Functional Response Criteria**	**PERCIST** [146]	**imPERCIST** [142]	**PERCIMT** [143]
Complete metabolic response (CMR)	Complete resolution of [18F]FDG uptake within all lesions to a level of less than or equal to that of the mean liver activity and that is indistinguishable from the background (blood pool uptake).	Complete resolution of [18F]FDG uptake within all lesions to a level of less than or equal to that of the mean liver activity and that is indistinguishable from the background (blood pool uptake).	Complete resolution of [18F]FDG uptake within all lesions to a level of less than or equal to that of the mean liver activity and that is indistinguishable from the background (blood pool uptake).
Partial metabolic response (PMR)	Reduction of at least 30% in the sum of SULpeak of all target lesions detected at baseline and an absolute drop of 0.8 SULpeak units.	Reduction of at least 30% in the sum of SULpeak of all target lesions detected at baseline and an absolute drop of 0.8 SULpeak units.	Reduction of at least 30% in the sum of SULpeak of all target lesions detected at baseline and an absolute drop of 0.8 SULpeak units.
Stable metabolic disease (SMD)	Does not meet the criteria for CR, PR, or PD.	Does not meet the criteria for CR, PR, or PD.	Does not meet the criteria for CR, PR, or PD.
Progressive metabolic disease (PMD)	Increase of at least 30% in the sum of SULpeak of all target lesions detected at baseline and an absolute increase of 0.8 SULpeak units. Or 75% increase in total lesions glycolysis (TLG) with no decrease in SUL. Or The appearance of one or more new FDG-avid lesions that are typical of cancer and that are not related to inflammation or infection is always considered progression.	Increase of at least 30% in the sum of SULpeak of all target lesions detected at baseline, or new FDG-avid lesions are considered UPMD and must be confirmed 4–8 weeks later as CPMD. Progression is not confirmed in case of PMR or SMD at 4–8 weeks, and evaluation must be reset.	Progressive disease if: ≥4 new lesions (<1 cm in functional diameter); ≥3 new lesions (>1 cm in functional diameter); ≥2 new lesions (>1.5 cm in functional diameter).

**Note**: *CPMD* = confirmed progressive metabolic disease; *iCPD* = immune confirmed progressive disease; *imPERCIST* = immunotherapy-modified PET Response Criteria in Solid Tumors; *imRECIST* = immunotherapy-modified Response Evaluation Criteria in Solid Tumors; *irRC* = immune-related response criteria; *iUPD* = immune unconfirmed progressive disease; *PERCIMT* = PET Response Evaluation Criteria for Immunotherapy; *PERCIST* = PET Response Criteria in Solid Tumors; *RECIST* = Response Evaluation Criteria in Solid Tumors; *UPMD* = unconfirmed progressive metabolic disease.

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
