# Peer review of "NSCLC Biomarkers to Predict Response to Immunotherapy with Checkpoint Inhibitors (ICI): From the Cells to In Vivo Images"

_cancers, 2021, doi:10.3390/cancers13184543_

Round 1
Reviewer 1 Report
The authors present a very extensive, but well-founded review article on predictive factors for immune checkpoint inhibitor therapy in NSCLC, including tumor- and patient-based as well as imaging biomarkers. Overall, the article appears appealing to me, but is a little lenghty to read.
I have some minor comments to make:
Abstract:
- line 30: I would regard immunotherapy using ICI as "targeted therapy", so that this sentence should be rephrased
- line 32/33: "TME" should be spelled out as "tumor microenvironment", but is "host immune microenvironment" in the text
Introduction:
- line 57: This sentence is somehow misleading: 25% of NSCLC may be oncogene-addicted, however most of these cannot be targeted by TKI at the moment
- Table 1: As far as I am informed, there is an EMA-opinion on Atezolizumab/bevacizumab/chemotherapy for the use in TKI-pre-treated ALK or EGFR-positive NSCLC? At least, this is one of the main indications where this combination treatment is being used at the moment. Could the authors provide more information on that issue?
In-vitro biomarkers:
- 285: A favourable effect, or: favourable effects
- 337: ... from small biopsies.
General remarks:
- The article is quite extensive and could benefit from some shortening. Especially, table 1 could from my point of view be omitted, as it does not contribute to the target topic of predictive biomarkers, but just lists approved ICI drugs.
- Also, the very extensive part on molecular imaging would benefit from some shortening. Even though the topic is interesting and has large future potential, it seems rather over-represented given the largely experimental and non-validated nature of the reported evidence.
- Also, there are some minor shortcomings in syntax and spelling, the authors should consider proofreading by an english native-speaker
Author Response
The authors present a very extensive, but well-founded review article on predictive factors for immune checkpoint inhibitor therapy in NSCLC, including tumor- and patient-based as well as imaging biomarkers. Overall, the article appears appealing to me, but is a little lenghty to read. I have some minor comments to make:
Abstract:
- line 30: I would regard immunotherapy using ICI as "targeted therapy", so that this sentence should be rephrased
A: We thank the Reviewer for the observation. Accordingly, for further clarity, we modified the revised manuscript using the expression “molecular targeted therapies”, to stress the contrast with the mechanism of action of checkpoint inhibitors.
- line 32/33: "TME" should be spelled out as "tumor microenvironment", but is "host immune microenvironment" in the text
A: Agreed. Amended as suggested.
Introduction:
- line 57: This sentence is somehow misleading: 25% of NSCLC may be oncogene-addicted, however most of these cannot be targeted by TKI at the moment
A: Agreed. Amended to be clearer.
- Table 1: As far as I am informed, there is an EMA-opinion on Atezolizumab/bevacizumab/chemotherapy for the use in TKI-pre-treated ALK or EGFR-positive NSCLC? At least, this is one of the main indications where this combination treatment is being used at the moment. Could the authors provide more information on that issue?
A: We thank the Reviewer for this comment and apologize for the missing indication. We integrated the atezolizumab indication in the revised Table1 of the manuscript.
In-vitro biomarkers:
- 285: A favourable effect, or: favourable effects
A: We made correction.
- 337: ... from small biopsies.
A: We made correction.
General remarks:
- The article is quite extensive and could benefit from some shortening. Especially, table 1 could from my point of view be omitted, as it does not contribute to the target topic of predictive biomarkers, but just lists approved ICI drugs.
- Also, the very extensive part on molecular imaging would benefit from some shortening. Even though the topic is interesting and has large future potential, it seems rather over-represented given the largely experimental and non-validated nature of the reported evidence.
A: We thank the Reviewer for her/his comment, we understand that the paper is quite long, on the other hand the topic we set out to cover was quite extensive. The review is addressed to different professional figures, so we believe that Table 1 can be an added value, allowing to better frame the use of immunotherapy in NSLC tumor (to our knowledge there is currently no table in the literature that summarizes this scheme of use). Even with regard to the imaging part, which is certainly very extensive, we find it challenging to reduce the information about the developing part on the use of radiomics and artificial intelligence, topics now preponderant in radiology and nuclear medicine fields. Moreover, we have to consider that, although widely used in clinical practice, there is currently no validation for the use of PET imaging in the evaluation of immunotherapy, so all this part can be considered experimental.
- Also, there are some minor shortcomings in syntax and spelling, the authors should consider proofreading by an English native-speaker
A: We thank the Reviewer for her/his comment, the paper has been viewed and corrected by a proofreading English native-speaker
Reviewer 2 Report
In this review, authors summarized the ongoing research on factors predictive for response to immune checkpoint inhibition (ICI) in lung cancer. The manuscript is well organized and of sure interest for the broad readership of Cancers.
In my opinion the review is suitable for publication, although a couple of things may be added for the sake of completeness:
- In paragraph 3.1.3, authors should add a reference to Koyama et al, Cancer Res 2016 (STK11/LKB1 Deficiency Promotes Neutrophil Recruitment and Proinflammatory Cytokine Production to Suppress T-cell Activity in the Lung Tumor Microenvironment) when they describe tumor genotypes linked to ICI resistance, in particular when they cite mutations in the STK11
- Somewhere in chapter 3 (maybe in paragraph 3.3) authors should add a reference regarding the correlation between circulating IL-8 levels and the non response to ICI (Sanmamed et al, Ann Oncol 2017 and/or Schalper et al, Nature medicine 2020).
- Authors should spend some word in the Introduction to cite another group of patients who undergo a rapid disease progression defined as “hyperprogression”, although no specific markers to identify such patients are known.
Author Response
In this review, authors summarized the ongoing research on factors predictive for response to immune checkpoint inhibition (ICI) in lung cancer. The manuscript is well organized and of sure interest for the broad readership of Cancers.
In my opinion the review is suitable for publication, although a couple of things may be added for the sake of completeness:
- In paragraph 3.1.3, authors should add a reference to Koyama et al, Cancer Res 2016 (STK11/LKB1 Deficiency Promotes Neutrophil Recruitment and Proinflammatory Cytokine Production to Suppress T-cell Activity in the Lung Tumor Microenvironment) when they describe tumor genotypes linked to ICI resistance, in particular when they cite mutations in the STK11
A: We thank the Reviewer for her/his valuable suggestion. In accordance we have integrated the relative section.
- Somewhere in chapter 3 (maybe in paragraph 3.3) authors should add a reference regarding the correlation between circulating IL-8 levels and the non response to ICI (Sanmamed et al, Ann Oncol 2017 and/or Schalper et al, Nature medicine 2020).
A: We thank the Reviewer for her/his valuable suggestion. In accordance, we have integrated the revised manuscript adding the evidence on IL-8 in the relative section 3.3.2, considering the production of IL-8 by neutrophils.
- Authors should spend some word in the Introduction to cite another group of patients who undergo a rapid disease progression defined as “hyperprogression”, although no specific markers to identify such patients are known.
A: We thank the Reviewer for this comment. We have added a short paragraph on “hyperprogression” in the opening section. The topic is then taken up in the nuclear medicine section
Reviewer 3 Report
The article is extremely complete is providing a state-of-the-art for IO therapy in lung cancer. Further the article is very well written and presented in a clear, logical progression.
The article is suitable for publication after concerns are addressed:
1) (Table 1) kindly provide references in lieu of links. Providing both may be a reasonable alternative.
2) There should be a section providing similar level of definition for circulating biomarkers...autoantibodies, protein factors, metabolomic markers, etc. This is the only missing piece to this work being 'complete'.
Author Response
The article is extremely complete is providing a state-of-the-art for IO therapy in lung cancer. Further the article is very well written and presented in a clear, logical progression.
The article is suitable for publication after concerns are addressed:
- (Table 1) kindly provide references in lieu of links. Providing both may be a reasonable alternative.
A: Agreed. Amended as suggested.
- There should be a section providing similar level of definition for circulating biomarkers...autoantibodies, protein factors, metabolomic markers, etc. This is the only missing piece to this work being 'complete'.
A: We thank the Reviewer for the valuable comment. As we structured the manuscript and divided the sections, circulating biomarkers are included in the “in-vitro” section and are described in the paragraphs according to the relationship with tumor, microenvironment, or host. As an example, description on blood-TMB is found in the 3.1.2 section (Tumor mutational burden). As well, CXCL13, since it is produced by CD8 T lymphocytes is found in the 3.2.2 section (3.2.2. Tumor infiltrating CD8+ T Cell: Phenotype and TCR clonality). In the revised manuscript, evidence on the possible predictive role of circulating IL-8 are also described, in the 3.3.2 section (Innate immune population).
Reviewer 4 Report
This review article is trying to evaluate the possible NSCLC biomarkers to predict response to immunotherapy with checkpoint inhibitors. The authours provide an overview on both in-vitro and in-vivo biomarkers, already in use or under development, that may enrich for clinical benefit in NSCLC patient’s candidate to ICIs treatment. The results are well organized. However, some typos need to be checked.
Majory Points:
- Please remove the “2. Materials and Methods” section and write part of the content in the introduction.
- In line 202, does the “(≥/<20 mutations per megabase), was negative” means that all the mutations is negative?
- The scheme or table summarizing each part would be of help. Please provide a graphic figure to summerize this review paper.
Minor Points:
Some text formats are wrong in line 578, 599, and 680, please correct with [18F]
Author Response
This review article is trying to evaluate the possible NSCLC biomarkers to predict response to immunotherapy with checkpoint inhibitors. The authours provide an overview on both in-vitro and in-vivo biomarkers, already in use or underdevelopment, that may enrich for clinical benefit in NSCLC patient’s candidate to ICIs treatment. The results are well-organized. However, some typos need to be checked.
Majory Points:
- Please remove the “2. Materials and Methods” section and write part of the content in the introduction.
A: Agreed. Amended as suggested.
- In line 202, does the “(≥/<20 mutations per megabase), was negative” means that all the mutations is negative?
A: We thank the Reviewer for her/his comment. In line 202 of the original manuscript “was negative” refers to the NEPTUNE trial. To improve clarity of the sentence, we modified the revised manuscript as follows: “the NEPTUNE trial, which prospectively assessed response to double ICIs (durvalumab plus tremelimumab) according to TMB level in blood, failed to demonstrate longer OS in patients with high TMB (≥20 mutations per megabase)”. This means that response to ICI in TMB low patients (those harboring tumors with somatic non-synonymous mutations below the threshold of 20 per magabase) was not significantly different from those with high TMB (over 20 mutations per megabase).
- The scheme or table summarizing each part would be of help. Please provide a graphic figure to summarize this review paper.
A: We thank the Reviewer for her/his valuable suggestion. In accordance, we have created a graphic figure to summarize this review paper.
Minor Points:
Some text formats are wrong in line 578, 599, and 680, please correct with [18F]
A: We made correction.